


# A global database on holdover time of lightning-ignited wildfires

Jose V. Moris[1], Pedro Álvarez-Álvarez[2], Marco Conedera[3], Annalie Dorph[4], Thomas D. Hessilt[5], Hugh G. P. Hunt[6], Renata Libonati[7], Lucas S. Menezes[7], Mortimer M. Müller[8], Francisco J. Pérez-Invernón[9], Gianni B. Pezzatti[3], Nicolau Pineda[10], Rebecca C. Scholten[5], Sander Veraverbeke[5], B. Mike Wotton[11], Davide Ascoli[1]

[1]Department of Agricultural, Forest and Food Sciences DISAFA, University of Turin, Grugliasco, Italy
[2]Department of Organisms and Systems Biology, University of Oviedo, Mieres, Spain
[3]Insubric Ecosystem Research Group, Swiss Federal Institute for Forest, Snow and Landscape Research WSL, Cadenazzo, Switzerland
[4]FLARE Wildfire Research, University of Melbourne, Creswick, Australia
[5]Faculty of Science, Vrije Universiteit Amsterdam, Amsterdam, the Netherlands
[6]Johannesburg Lightning Research Laboratory, University of the Witwatersrand, Johannesburg, South Africa
[7]Departament of Meteorology, Federal University of Rio de Janeiro, Rio de Janeiro, Brazil
[8]Institute of Silviculture, University of Natural Resources and Life Sciences BOKU, Vienna, Austria
[9]Institute of Astrophysics of Andalusia IAA-CSIC, Granada, Spain
[10]Meteorological Service of Catalonia, Barcelona, Spain
[11]Canadian Forest Service, Sault Ste. Marie, Canada

*Correspondence to*: Jose V. Moris (moris.josev@gmail.com)

**Abstract.** Holdover fires are usually associated with lightning-ignited wildfires (LIWs), which can experience a smouldering phase or go undetected for several hours to days and weeks before being reported. Since the existence and duration of the smouldering combustion in LIWs is usually unknown, holdover time is conventionally defined as the time between the lightning event that ignited the fire and the time the fire is detected. Therefore, all LIWs have an associated holdover time, which may range from a few minutes to several days. However, we lack a comprehensive understanding of holdover times. Here, we introduce a global database on holdover times of LIWs. We have collected holdover time data from 29 different studies across the world through a literature review and datasets assembled by authors of the original studies. The database is composed of three data files (censored data, non-censored data, ancillary data) and three metadata files (description of database variables, list of references, reproducible examples). Censored data are the core of the database and consist of different frequency distributions reporting the number or relative frequency of LIWs per interval of holdover time. In addition, ancillary data provide further information to understand the methods and contexts in which the data were generated in the original studies. The first version of the database contains 42 frequency distributions of holdover time built with data on more than 152,375 LIWs from 13 countries in five continents covering a time span from 1921 to 2020. This database is the first freely available, harmonized, and ready-to-use global source of holdover time data, which may be used in different ways to investigate LIWs and model the holdover phenomenon. The complete database can be downloaded at https://doi.org/10.5281/zenodo.7352172 (Moris et al., 2022).



## 1 Introduction

Lightning-ignited wildfires (LIWs) are a major component of fire regimes in remote and mountainous regions (e.g., Hanes et al., 2019; Moris et al., 2020). Climate change is expected to increase the frequency and burned area of LIWs in certain regions (Hessilt et al., 2022), which in turn may affect the carbon cycle (Chen et al., 2021). There are still important knowledge gaps about LIWs. Whereas LIWs are often studied in boreal and temperate ecosystems of North America (e.g., Abatzoglou et al., 2016; Veraverbeke et al., 2017), in other regions such as Europe and Australia, LIWs receive less attention because of their lower occurrence or burned area in comparison with human-caused fires (Conedera et al., 2006; Ganteaume et al., 2013; Ganteaume and Syphard, 2018; Dorph et al., 2022). Similarly, LIWs are less studied in South America, Asia and Africa (e.g., Manry and Knight, 1986; Kharyutkina et al., 2022; Menezes et al., 2022).

The physical process involved in LIWs is commonly divided into three phases: ignition, survival and arrival (Anderson, 2002; Pineda and Rigo, 2017). The ignition is caused by a cloud-to-ground (CG) lightning strike. We know little about which fuels are more frequently first ignited by lightning, although the organic soil layers surrounding the base of trees hit by lightning are reported to be a common ignition point (Plummer, 1912; Taylor, 1969; Ogilvie, 1989). The survival phase, which refers to smouldering combustion, does not occur in every LIW (Cesti et al., 2005). Depending on environmental conditions (e.g., fuel dryness and weather), the ignition may spread almost immediately as a surface fire or, alternatively, survive as a smouldering fire in the organic soils (Anderson et al., 2000; Martell and Sun, 2008). Therefore, the survival phase, also known as smouldering or holdover phase, is characterized by the smouldering combustion (i.e., slow, low temperature, flameless burning) of the soil organic layers (Rein, 2016). It is assumed that the rain and weather conditions associated with thunderstorms are usually unfavorable to sustain flaming combustion (Pérez-Invernón et al., 2021; Soler et al., 2021). This may result in a smouldering phase within the litter, duff or humus layers until LIWs extinguish themselves or conditions become more favorable (e.g., drying out of surface fine fuels or strong winds) for a transition to flaming combustion (Show and Kotok, 1923; Taylor, 1969; Anderson et al., 2000; Pineda and Rigo, 2017). When a LIW reaches the final arrival phase, or flaming combustion, the faster spread and higher energy and smoke released by a surface fire facilitate its detection. Survival and arrival phases may also alternate during a LIW because flaming combustion lapses back into the survival phase driven by irregular and changing conditions of fuels and weather, e.g., overnight (Anderson et al., 2000; Anderson, 2002; Cesti et al., 2005). Furthermore, it is assumed that changes in environmental conditions can extinguish some LIWs during the survival phase before being detected and reported (Anderson et al., 2000; Wotton and Martell, 2005; Dowdy and Mills, 2009).

Since LIWs occur often in remote areas, the processes and behavior of smouldering wildfires are difficult to study. As a result, these LIWs, which are commonly referred to as "holdover fires" in the scientific literature, remain poorly studied (Rein and Huang, 2021). A holdover fire may refer to any wildfire, human- or lightning-caused, with a smouldering phase



that remains undetected for a considerable time, including overwintering fires (Scholten et al., 2021). However, holdover fires are usually associated with LIWs that experience the survival phase (Flannigan and Wotton, 1991; Schultz et al., 2019),

or simply LIWs that go undetected for an arbitrary duration, such as several hours or days (Show and Kotok, 1923; Taylor, 1969; Anderson, 2002). Given that the existence and duration of the survival phase is usually unknown, "holdover time" is conventionally defined, for practical reasons, as the time between lightning-induced fire ignition and fire detection (Wotton and Martell, 2005; Dowdy and Mills, 2009; Braun and Stafford, 2016). According to this definition, all LIWs have an associated holdover time, and holdover times may range from a few minutes (e.g., Pineda and Rigo, 2017) to several days,

and occasionally some weeks and even months (Frost et al., 2018). A typical example of LIWs with a survival phase are evening and night ignitions that smoulder overnight and are detected the day after in the afternoon when a higher temperature and lower relative humidity favor fire spread (Pineda et al., 2014; Pineda and Rigo, 2017). While it is commonly accepted that the majority of LIWs have short holdover times (e.g., < 24 hours; Dowdy and Mills, 2009; Schultz et al., 2019; Moris et al., 2020; Pineda et al., 2022), relative frequencies of LIWs with longer holdover durations are less generalizable,

partially due to limited understanding on how often and how long LIWs can smoulder (Scholten et al., 2021). However, most studies show that holdover time follows a right-skewed distribution, with an exponential-like decay with increasing time (e.g., Nash and Johnson, 1996; Wotton and Martell, 2005; Schultz et al., 2019; Moris et al., 2020).

Most data on holdover time during the 20th century come from individual fire reports collected by forest authorities (e.g.,

Kourtz, 1967; Barrows, 1951; Barrows, 1978). In forests of the western United States, observers stationed at fire lookouts not only reported wildfires, but also information on storm characteristics (Gisbone, 1926; Gisbone, 1931). For each visible LIW, holdover time was calculated as the time elapsed between the discovery of the wildfire and the most recent lightning storm reported over the concerned area (Gisbone, 1926; Morris, 1947). With the development of modern ground-based Lightning Location Systems (LLS; Cummins and Murphy, 2009), holdover times began to be estimated by matching wildfire

and lightning data from LLS (Nash and Johnson, 1996; Wotton and Martell, 2005). Unfortunately, due to data inaccuracies in combination with holdover times, usually we cannot unambiguously distinguish the lightning strike that ignited a wildfire. Therefore, several lightning events, close enough in time and space to the reported wildfire, may be indicated as possible candidates for the ignition source (Dowdy and Mills, 2009; Braun and Stafford, 2016; Moris et al., 2020). Accordingly, methods developed to match wildfires and lightning rather search for the most likely individual lightning event that ignited

the wildfire (Larjavaara et al., 2005; Pineda et al., 2014). Current methods apply a buffer area centered at the LIW ignition point to account for location errors of both lightning and wildfires, and a temporal window backward from the LIW discovery time to account for holdover time (Moris et al., 2020). Holdover time is then calculated as the time between the strike of the most probable lightning (i.e., the time of ignition recorded by the LLS) and the LIW discovery time (e.g., reported by a fire database).

Despite the importance of the holdover phenomenon to understand the initial behavior of LIWs and identify lightning events causing wildfires, we lack a synthesis on the variability of holdover times, as well as any type of data source on holdover time that can be used for practical applications (e.g., data modeling). In this paper, we present the construction and structure of a global database of LIW holdover times with the aim of making these broad, harmonized and ready-to-use data on holdover time freely available to the community. The core of the database consists of frequency distributions of holdover times collected from numerous studies carried out during the last century in different regions, as well as metadata useful to understand the context of each dataset.

## 2 Methodology

### 2.1 Literature search and data sources

We conducted literature searches to identify potential sources of holdover time data using academic databases and search engines: Scopus, Web of Science, JSTOR, ScienceDirect, SpringerLink, Scilit, Google Scholar, AGRIS, Canadian Forest Service Publications, and Treesearch. We used the search terms "lightning fire" and "holdover fire", with emphasis on the title, abstract and keywords. The initial screening for relevant documents focused on figures, tables, and the presence of specific key words in the texts (i.e., holdover, latent, smoulder, survival, phase, elapse, time, detection, discovery, and lightning). Once an initial set of relevant publications were identified, we read them carefully to find holdover time data and additional information regarding how these data were obtained. During this phase, we found other potential data sources within the references of these publications.

We identified 35 studies with potential data on holdover time. A few studies were discarded because the data were repeated or could not be extracted. For studies not showing details on holdover times, we contacted the corresponding authors to request for data. We also contacted corresponding authors of studies carried out from 2020 onwards to request for the original holdover time data. We ended up collecting data on holdover durations from 29 different studies across the world.

### 2.2 Data collection

According to the available information, three kinds of data were acquired: censored data on holdover time, non-censored data on holdover time, and ancillary data. Censored data are the core of the database and consist of frequency distributions. Frequency distributions report the number or relative frequency of LIWs for which we do not know the exact holdover times but the lower and upper limits of the time interval surrounding the holdover times (i.e., interval-censored data). Right-censored data (i.e., only the lower limit is known) were included rarely. Some censored datasets were provided by authors of the original studies, and the rest were collected from figures, tables, texts, appendices, and unpublished records from the sources identified in Sect. 2.1. We used the WebPlotDigitizer tool to extract data values from figures (Rohatgi, 2021). We

did not set a minimum number of LIWs, but all frequency distributions included two or more time intervals. We compiled more than one frequency distribution from some studies in which different study areas were analyzed separately.

Non-censored data refer to estimated values of continuous holdover time without any censoring (i.e., the exact estimated
value of holdover time for each single LIW). Datasets of non-censored data were compiled by authors of the original studies and were also used to build some of the frequency distributions included in the censored data.

Finally, we collected data describing and summarizing the studies from which the holdover times were estimated. These ancillary data contain information related to spatial, temporal, methodological, fire, and lightning aspects extracted from the
original studies, which are important to understand the methods and contexts in which the data were generated. Additional external data sources were used to obtain information on the main biomes (Olson et al., 2001) and climate classes (Beck et al., 2018) of the study areas.

## 2.3 Data harmonization and quality control

The original data were not presented consistently across the various data sources. Consequently, some data variables were
harmonized to facilitate the comparison of frequency distributions and studies. Most of the harmonization process consisted of assigning classes to ancillary data, and reporting the same units for all values of a variable. For instance, we standardized the time interval bounds of frequency data by reporting all times in days and starting at day zero. Few datasets reported negative values of holdover time (i.e., fire detections were reported before estimated ignition times) because of temporal uncertainties in the ignition data. Those particular fires were included within the first time interval of the frequency
distributions (i.e., we assumed short holdover times for those LIWs) to solve this inconsistency.

We double-checked for errors and inconsistencies within the database. The data were first checked automatically in R (R core team, 2021). We manually inspected all records (rows) for all variables (columns) of censored, non-censored, and ancillary data. The data provided by authors of original studies were also verified by the same authors. In the database, null
values are not strictly reserved for variables where the required information is not applicable. Occasionally, we were unable to obtain some data. For example, in some frequency distributions of holdover time we collected data on relative frequencies but not on number of fires.

The database may include some duplicate data. For instance, certain LIWs may be used in more than one dataset (i.e.,
frequency distribution). This is likely in studies with overlapping study areas and years, especially when the same fire data sources were used by the same authors. However, we did not attempt to correct this for two reasons. First, it was not possible to identify "duplicate data" because of the data aggregation in the original studies. Generally, censored data were collected at study area level, and consequently the coordinates and times of ignition and discovery of single LIWs could not be retrieved.



Second, we believe that overlaps do not imply the presence of redundant data given that each dataset included in the

database is unique over a particular study area and time period.

## 2.4 Data description

The database on holdover time of LIWs is composed of three Comma-Separated Values (CSV) data files (censored data, non-censored data, ancillary data), and three complementary HyperText Markup Language (HTML) metadata files that support the data files (description of database variables, list of references, reproducible examples). For each data record

(row) of a frequency distribution (dataset) in the censored data file, we provided twelve variables (columns), which are described in Table 1. The rows of censored data correspond to the time intervals in which the frequency distribution of holdover time are divided (Fig. 1). The duration of these time intervals can vary substantially between datasets (Fig. 1), and also within the same frequency distribution. When data on number of LIWs per time interval were not available, only relative frequencies were provided instead. Regarding non-censored data, all values of holdover time are reported in hours.


**Table 1.** Overview of the variables of censored data from the database on holdover time of LIWs.

| Variable | Description |
|---|---|
| Study_id | ID code referring to the original study and dataset. |
| Reference | In-text citation of the original study. |
| Time_interval | Duration of the time interval. |
| Time_interval_d | Duration of the time interval in days. |
| Lower_limit_d | Lower bound of the time interval in days. |
| Upper_limit_d | Upper bound of the time interval in days. |
| N_fires | Number of LIWs with an estimated holdover time within the time interval. |
| RF | Relative frequency of LIWs in the time interval. |
| CRF | Cumulative relative frequency of LIWs. |
| Original_data | How the frequency distribution was reported in the original study (N = number of fires; P = relative frequency). |
| Data_location | Where the data were reported within the original study. |
| Collection_method | Method used to collect the data (Copied from original; WebPlotDigitizer; Personal communication). |

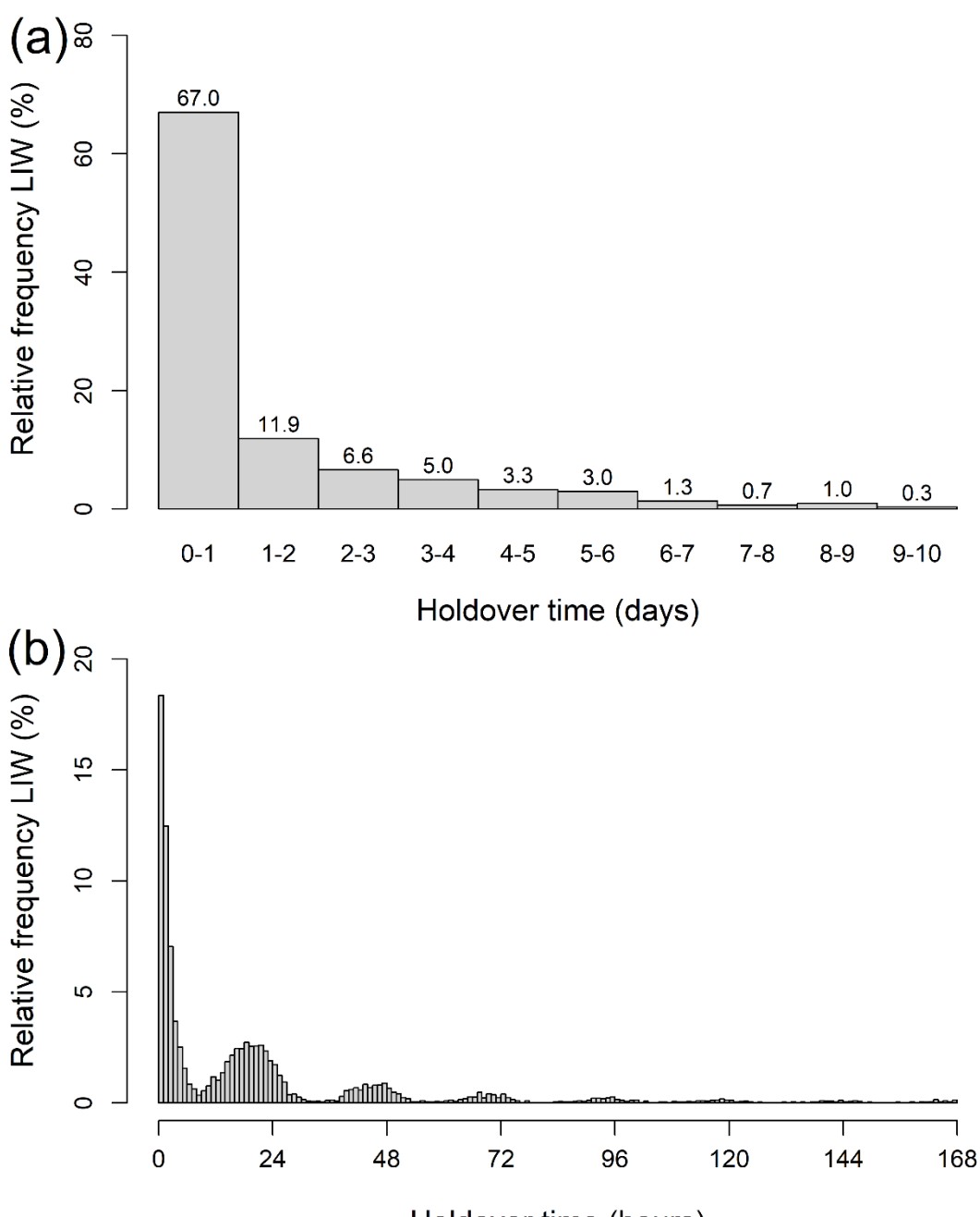

**Figure 1.** Examples of frequency distributions (i.e., censored datasets) of LIW holdover time from the database. (a) Daily
frequency distribution obtained from 303 LIWs occurred in Austria between 2013 and 2020 (Müller and Vacik, 2017). (b)
Hourly frequency distribution obtained from 6301 LIWs occurred in Arizona and New Mexico between 2009 and 2013
(Pérez-Invernón et al., 2022).



Ancillary data were organized into 33 variables (columns). These variables were grouped into seven major groups to
facilitate the description of the information (Table 2): data identification, spatial, temporal, fire, lightning, methodology, and
data entry variables. Each row of the ancillary data represents a frequency distribution from the censored data and is
identified with a unique code. A complete description of all the variables of the database is included in a separate file.
Moreover, another file includes a list of references with the full citation of all the original data sources used to build the
database, while the last file includes some reproducible examples for using the data within the R statistical environment.


**Table 2.** Overview of the variables of ancillary data from the database on holdover time of LIWs.

| Variable | Description |
|---|---|
| Variables on data identification: | |
| Study_id | ID code referring to the original study and dataset. |
| Reference | In-text citation of the original study. |
| Type_publication | Type of original study (Paper; Proceeding; Report; Thesis; Unpublished). |
| Variables on spatial data: | |
| Study_area | Territory in which the LIWs were studied. |
| Country | Country of the study area. |
| ISO_code | Country code or region code of the study area according to ISO 3166. |
| Spatial_scale | Spatial extent of the study area (Local; Regional; Continental; Global). |
| Biome | Most common biome of the study area according to Olson et al. (2001). |
| Ecozone | Biogeographic realm of the study area according to Olson et al. (2001). |
| Climate_class | Most common climate class of the study area according to the Köppen-Geiger climate classification by Beck et al. (2018). |
| Variables on temporal data: | |
| Start_year | Starting year of the study period. |
| End_year | Ending year of the study period. |
| Length_year | Length of the study period in years. |
| Min_time_h | Minimum time interval in hours of the censored data. |
| Max_time_h | Maximum time interval in hours of the censored data. |
| Variables on fire data: | |
| Number_fires | Total number of LIWs for which the holdover times were estimated. |
| Number_records | Total number of time intervals in which the frequency distribution of holdover time data was divided. |
| Fire_detection | Source of wildfire discovery data (Fire database; Remote sensing). |



| | |
|---|---|
| Fire_data_source | Dataset used to extract LIW data. |
| Variables on lightning data: | |
| LLS | Lightning Location System (LLS) used to obtain lightning data. |
| Lightning_level | Level at which lightning data were matched with wildfire data (Stroke; Flash). |
| DE_pct | Detection Efficiency (DE) refers of the expected percentage of lightning discharges reported by the LLS. |
| LA_km | Location Accuracy (LA) in km usually refers to the expected median error between the reported CG stroke locations by the LLS and the real ground strike points. |
| Variables on methodology: | |
| Method | Methodological approach used to estimate holdover times of LIWs (Storm time; Lightning match). |
| Buffer_distance_km | Maximum buffer radius in km around the wildfire ignition point used to select potential igniting lightning. |
| Temporal_window_d | Maximum temporal window backward in days from the wildfire detection time used to select potential igniting lightning. |
| Max_holdover_d | Maximum estimated holdover time in days. |
| Selection_criteria | Criteria used to select the most likely igniting lightning (Minimum holdover time; Daily minimum distance; Maximum proximity index; Decision tree). |
| Variables on data entry: | |
| Dataset | Reference number used to distinguish the holdover time distribution in case different methods were applied to the same dataset. |
| Data_collector | Person who filled the data records. |
| Date_entry | Date on which the data records were filled. |
| Data_check | Whether or not the data records were double-checked by a different person from the one who filled the data records (Yes; No). |
| Comments | Additional notes about the data or original study. |

## 3 Overview of contents

The database contains 42 frequency distributions of censored holdover time data (Table 3) and nine non-censored datasets of single fire-level holdover time (Table 4). Individual time intervals of censored data go from one minute to 87 days, although hourly and daily intervals are the most frequent durations (Fig. 1). Censored data come from 29 different studies, mostly published in peer-reviewed journals, representing five major vegetated biomes (Fig. 2), and distributed across 13 countries in five continents (Fig. 3): North America (United States and Canada), South America (Brazil), Europe (Spain, Italy,




Switzerland, Austria, France, Portugal, Greece and Finland), Asia (Russia) and Oceania (Australia). The studies cover a time span of a century (from 1921 to 2020), with diverse study periods lasting from one to 24 years (Fig. 3). In total, the database

includes 2,311 records of censored data obtained from more than 152,375 LIWs (Table 3). Frequency distributions were built with a variable number of LIWs (between 25 and 28,377), and 59.5% of the distributions exceed 500 LIWs (Table 3). Regarding the methodology to derive holdover times, 28.6% of the frequency distributions (all from the 20th century) used the elapsed time between discovery of the LIW and the most recent lightning storm over the area of ignition. The rest of frequency distributions (71.4%), from the late 20th and early 21th century, used lightning data from LLS. The maximum

proximity index (in 17 frequency distributions) and the minimum holdover time (in 10 frequency distributions) are the most recurrent criteria applied to select igniting lightning.

**Table 3.** Summary of the censored data from the database on holdover time of LIWs.

| Study id | Study area | Biome | Study period | Number fires | Number records | Median HOT (h) | Maximum HOT (d) | CRF d 1 (%) |
|---|---|---|---|---|---|---|---|---|
| SHO1923US01 | California (US) | Temperate coniferous forests | 1921–1921 | | 6 | 15.3 | | 67.0 |
| SHO1930US01 | California (US) | Temperate coniferous forests | 1921–1922 | 443 | 6 | 12.8 | | 68.0 |
| GIS1926US01 | Northern Rocky Mountains (US) | Temperate coniferous forests | 1924–1925 | 1933 | 11 | 4.8 | | 85.0 |
| GIS1931US01 | Northern Rocky Mountains (US) | Temperate coniferous forests | 1924–1928 | 4149 | 11 | 4.0 | | 86.0 |
| BAR1951US01 | Northern Rocky Mountains (US) | Temperate coniferous forests | 1931–1945 | 16368 | 13 | 4.2 | | 79.0 |
| MOR1948US01 | Oregon and Washington (US) | Temperate coniferous forests | 1940–1944 | 5357 | 28 | 6.4 | | 78.5 |
| TAY1969US01 | Northern Rocky | Temperate | 1950– | 14489 | 4 | 10.2 | | 77.0 |





| | | | | | | | | |
|---|---|---|---|---|---|---|---|---|
| | Mountains (US) | coniferous forests | 1965 | | | | | |
| KOU1967CA01 | Canada | Boreal forests | 1960–1963 | 3615 | 16 | 18.4 | | 59.6 |
| BAR1978US01 | Arizona and New Mexico (US) | Temperate coniferous forests | 1960–1974 | 28377 | 8 | 3.0 | | 90.2 |
| CON2006CH01 | Ticino (CH) | Temperate coniferous forests | 1981–2004 | 154 | 7 | 15.8 | 7.0 | 76.0 |
| DUN2010US01 | Florida (US) | Temperate coniferous forests | 1986–2003 | 230 | 2 | 23.4 | 23.0 | 51.3 |
| NAS1996CA01 | Alberta and Saskatchewan (CA) | Boreal forests | 1988–1993 | 2551 | 15 | 27.3 | 15.0 | 47.8 |
| WOT2005CA01 | Ontario (CA) | Boreal forests | 1992–2001 | 5169 | 28 | 44.5 | 28.0 | 33.5 |
| LAR2005FI01 | Finland | Boreal forests | 1996–2002 | 106 | 5 | 34.7 | | 42.5 |
| DOW2009AU01 | Victoria (AU) | Temperate broadleaf and mixed forests | 2000–2009 | 1797 | 4 | 18.5 | 90.0 | 64.7 |
| WOT2022CA01 | Boreal British Columbia (CA) | Boreal forests | 2000–2020 | 1393 | 22 | 21.2 | 22.0 | 56.6 |
| WOT2022CA04 | Saskatchewan (CA) | Boreal forests | 2000–2020 | 2983 | 22 | 24.9 | 22.0 | 49.3 |
| WOT2022CA03 | Alberta (CA) | Boreal forests | 2000–2020 | 10544 | 22 | 18.1 | 22.0 | 66.5 |
| WOT2022CA02 | Southern and Central British Columbia (CA) | Temperate coniferous forests | 2000–2020 | 16940 | 22 | 19.0 | 22.0 | 63.2 |
| HES2022US01 | Alaska (US) | Boreal forests | 2001– | 402 | 5 | 38.6 | 5.0 | 25.9 |

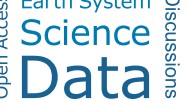



| | | | 2012 | | | | | |
|---|---|---|---|---|---|---|---|---|
| MOR2020CH01 | Switzerland | Temperate coniferous forests | 2001–2018 | 263 | 238 | 13.1 | 9.9 | 63.1 |
| HES2022CA01 | Northwest Territories (CA) | Boreal forests | 2001–2018 | 550 | 5 | 47.7 | 5.0 | 23.1 |
| WOT2022CA06 | Ontario (CA) | Boreal forests | 2001–2019 | 8005 | 22 | 53.6 | 22.0 | 25.5 |
| CON2006IT01 | Aosta Valley (IT) | Temperate coniferous forests | 2003–2003 | 25 | 6 | 20.0 | 6.0 | 60.0 |
| PIN2022ES01 | Catalonia (ES) | Mediterranean forests | 2003–2020 | 1013 | 233 | 1.8 | 9.7 | 84.4 |
| PIN2014ES01 | Catalonia (ES) | Mediterranean forests | 2004–2009 | 464 | 24 | 1.4 | 3.0 | |
| DOR2022AU01 | Victoria (AU) | Temperate broadleaf and mixed forests | 2004–2019 | 6777 | 120 | 1.5 | 5.0 | 87.4 |
| WOT2022CA05 | Manitoba (CA) | Boreal forests | 2004–2020 | 2542 | 22 | 39.8 | 22.0 | 38.8 |
| PER2022US02 | Florida (US) | Temperate coniferous forests | 2009–2013 | 2693 | 167 | 13.1 | 7.0 | 74.3 |
| PER2022US01 | Arizona and New Mexico (US) | Temperate coniferous forests | 2009–2013 | 6301 | 168 | 12.1 | 7.0 | 75.9 |
| PIN2017ES01 | Catalonia (ES) | Mediterranean forests | 2009–2014 | 357 | 19 | 1.6 | | 87.1 |
| PER2021PT01 | Portugal | Mediterranean forests | 2009–2015 | 309 | 93 | 15.9 | 3.9 | 64.1 |
| PER2021ES01 | Spain | Mediterranean forests | 2009–2015 | 2702 | 336 | 5.7 | 14.0 | 72.8 |
| PER2021FR01 | Mediterranean | Mediterranean | 2012– | 36 | 242 | 3.0 | 10.1 | 75.0 |



| | | | | | | | | |
|---|---|---|---|---|---|---|---|---|
| | France | forests | 2015 | | | | | |
| SCH2019US01 | United States | Temperate coniferous forests | 2012–2015 | 797 | 15 | 17.6 | 15.0 | 68.0 |
| MEN2022BR01 | Pantanal (BR) | Flooded grasslands and savannas | 2012–2017 | 265 | 65 | 21.6 | 2.7 | 61.5 |
| MOR2020IT01 | Aosta Valley (IT) | Temperate coniferous forests | 2012–2018 | 32 | 150 | 6.0 | 6.3 | 71.9 |
| HES2022US02 | Alaska (US) | Boreal forests | 2012–2018 | 287 | 5 | 37.0 | 5.0 | 28.2 |
| XUW2022RU01 | Yakutia (RU) | Boreal forests | 2012–2020 | 645 | 8 | 38.7 | 8.0 | 30.5 |
| MUL2021AT01 | Austria | Temperate coniferous forests | 2013–2020 | 303 | 10 | 17.9 | 10.0 | 67.0 |
| MAC2019US01 | Western United States | Temperate coniferous forests | 2017–2017 | 95 | 11 | 45.4 | 11.0 | 27.4 |
| PER2021GR01 | Greece | Mediterranean forests | 2017–2019 | 914 | 95 | 29.2 | 4.0 | 43.4 |

HOT = holdover time; CRF 1 d = cumulative relative frequency of LIWs with holdover time ≤ 24 hours; AU = Australia;

BR = Brazil; CA = Canada; CH= Switzerland; ES = Spain; IT = Italy; RU = Russia; US = United States.



**Table 4.** Summary of the non-censored data from the database on holdover time of LIWs.

| Study id | Study area | Biome | Study period | Number fires | Median HOT (h) | Minimum HOT (min) | Maximum HOT (d) | CRF d 1 (%) |
|---|---|---|---|---|---|---|---|---|
| MOR2020CH01 | Switzerland | Temperate coniferous forests | 2001–2018 | 263 | 13.0 | 1.1 | 9.9 | 63.1 |
| PIN2022ES01 | Catalonia (ES) | Mediterranean forests | 2003–2020 | 1013 | 1.7 | 0.6 | 9.7 | 84.4 |
| PER2022US02 | Florida (US) | Temperate coniferous forests | 2009–2013 | 2693 | 13.1 | 0.0 | 7.0 | 74.3 |
| PER2022US01 | Arizona and New Mexico (US) | Temperate coniferous forests | 2009–2013 | 6301 | 12.1 | 0.0 | 7.0 | 75.9 |
| PER2021PT01 | Portugal | Mediterranean forests | 2009–2015 | 309 | 15.6 | 1.0 | 3.8 | 64.1 |
| PER2021ES01 | Spain | Mediterranean forests | 2009–2015 | 2702 | 5.7 | 0.0 | 14.0 | 72.8 |
| PER2021FR01 | Mediterranean France | Mediterranean forests | 2012–2015 | 36 | 3.7 | 8.0 | 10.1 | 75.0 |
| MEN2022BR01 | Pantanal (BR) | Flooded grasslands and savannas | 2012–2017 | 265 | 21.7 | 1.0 | 2.7 | 61.5 |
| MOR2020IT01 | Aosta Valley (IT) | Temperate coniferous forests | 2012–2018 | 32 | 6.3 | 37.7 | 6.2 | 71.9 |

HOT = holdover time; CRF 1 d = cumulative relative frequency of LIWs with holdover time ≤ 24 hours; BR = Brazil; ES = Spain; IT = Italy; US = United States.






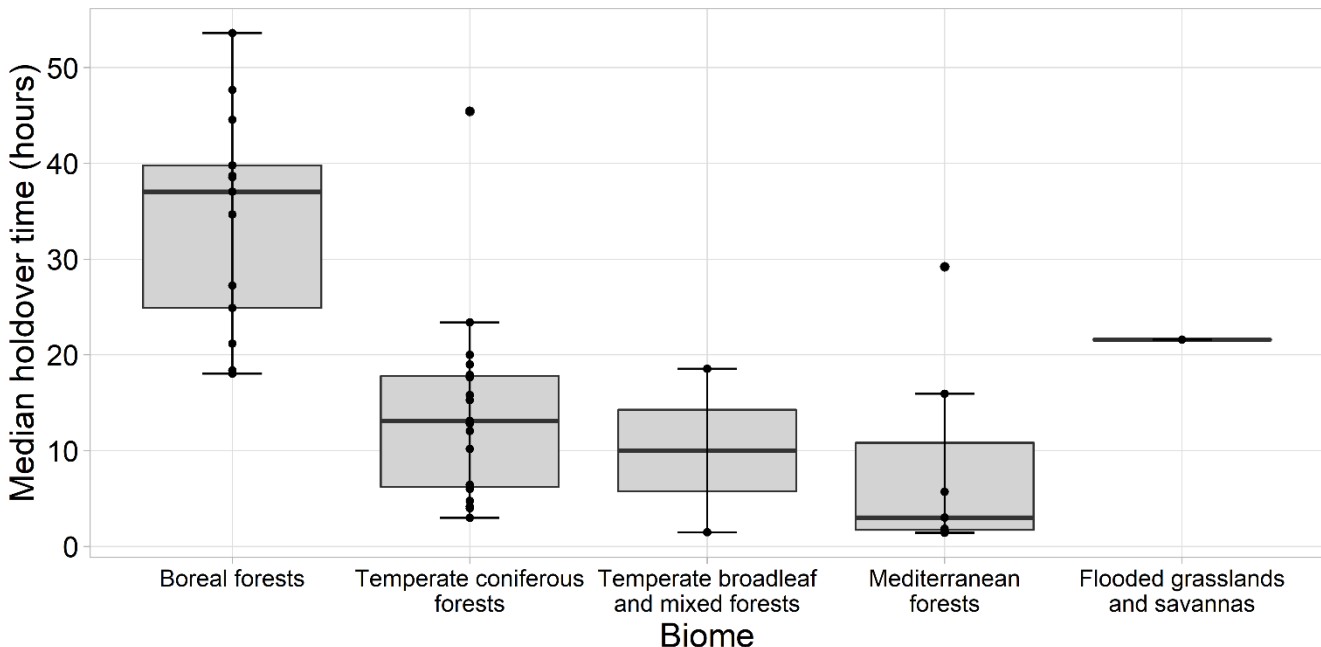

**Figure 2.** Boxplots of median values of holdover time by biome calculated from the 42 frequency distributions of censored data (Table 3).


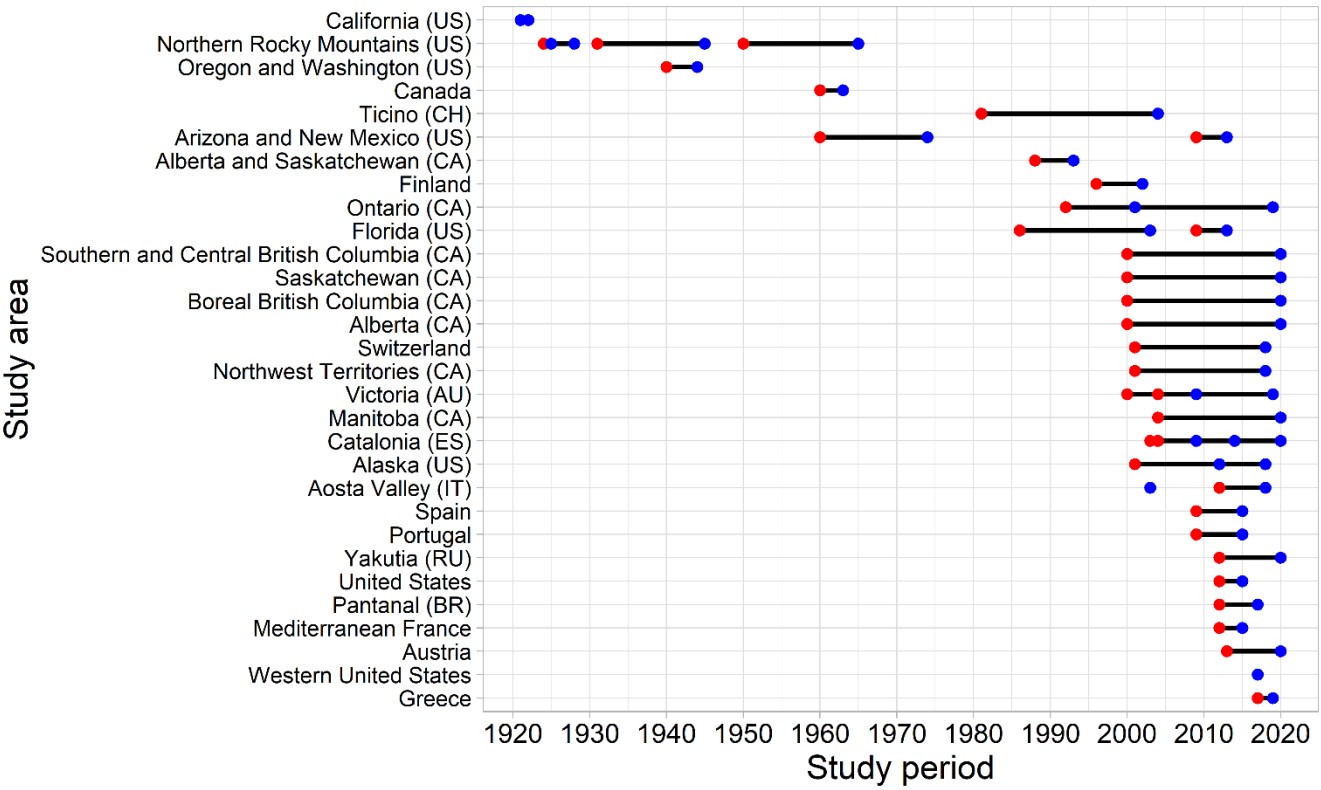

**Figure 3.** Study areas and periods included in the database on holdover time of LIWs. Red dots represent the starting years of the study periods in each study area, blue dots the ending years, and solid black lines the duration of the study periods. AU = Australia; BR = Brazil; CA = Canada; CH= Switzerland; ES = Spain; IT = Italy; RU = Russia; US = United States.


Censored and non-censored data show that the datasets of holdover time present right-skewed distributions (Fig. 1), with median holdover durations ranging from 1.4 to 53.6 hours (Table 3). Median values calculated from frequency data are influenced by the duration of the time intervals, and as a result median values calculated from non-censored data and hourly frequency distributions (i.e., regular 1-hour interval-censored frequencies) are more accurate than, for instance, values

calculated from daily distributions (i.e., regular 1-day interval-censored frequencies). Daily frequency distributions indicate that the first 24 hours are the most frequent interval of holdover time (e.g., Fig. 1a), with the exception of four distributions using fire remote sensing data from boreal regions ("HES2022CA01", "HES2022US01", "HES2022US02", "XUW2022RU01"), in which the second day (i.e., 24-48 hours) is the most frequent interval. In fact, hourly frequency distributions also illustrate that the first hour is the most frequent interval of holdover time (e.g., Fig. 1b), except for two

distributions ("MOR2020IT01", which is based on a low number of LIWs, and "MEN2022BR01", which uses fire remote sensing data). Censored data show that in 30 out of 42 distributions, the majority of LIWs (i.e., > 50% of LIWs) display holdover times of less than 24 hours, although the percentages of LIWs with holdover time below 24 hours vary between



datasets (Table 3). Finally, the maximum holdover times reported in the database are strongly influenced by the temporal thresholds applied in each original study, and may therefore not be good indicators of the maximum holdover times that
occur in the study areas (Table 3).

## 4 Discussion

The construction of the database faced several challenges that could be partially solved. Consequently, users of the present database should be aware of existing limitations. First and despite the long time span beginning in the 1920s (Show and Kotok, 1923; Gisbone, 1926; Show and Kotok, 1930; Gisbone, 1931), data on holdover times of LIWs are often hard to find
and relatively scarce in the scientific literature. Second, holdover time data are fragmented across different types of publications, and highly focused on forest ecosystems of North America and Europe. Third, some researchers did not include holdover time data in the original studies and the data remained unpublished. In other cases, holdover time data appear embedded in figures and may not be extracted accurately. Fourth, the database does not include coordinates and dates of wildfires and lightning. Consequently, the database cannot be used to study holdover times of individual LIWs in full detail
(e.g., Pineda et al., 2022). In that case, users of the database may contact the authors of the original studies, although often original data on wildfires and lightning may be difficult to share due to data privacy policies. Lastly, we did not search for scientific literature in other languages than English. In addition, we are aware that the database is not fully comprehensive and some holdover time datasets are not included in the current version of the database (e.g., Chen et al., 2015; Nampak et al., 2021). We therefore welcome any suggestion on other existing datasets that could be added to the database later on.


Frequency data show that the number of LIWs tends to decrease with increasing holdover time (Fig. 1a). Yet, hourly frequency distributions of holdover time that contain nearly 1,000 or more LIWs suggest the presence of daily cycles of holdover time (i.e., peaks or local maxima in the frequency of LIWs separate by approximately 24 hours; Fig. 1b). Morris (1947) already noticed the existence of local maxima 24 hours apart from each other. Pineda et al. (2022) illustrated how
these peaks are most likely associated with the diurnal heating and cooling cycle. The frequency distributions of both LIW detections and ignitions seem to follow bell-shaped distributions, with maximum values occurring between the late afternoon and early evening (i.e., 4-7 p.m.; Barrows, 1951; Pineda et al. 2022). Therefore, LIWs that smoulder for one or more nights may be more likely to be reported in the afternoon hours, when the environmental conditions become progressively warmer and drier, favoring a transition to flaming combustion.


Empirical holdover time distributions may also mask some unsolved methodological issues. For instance, we generally lack field observations of the ignition, survival and arrival phases of LIWs (Fuquay et al., 1967; Ogilvie, 1989; Rein, 2016; Santoso et al., 2019). Consequently, modern methods applied to identify igniting lightning, including the maximum proximity index (Larjavaara et al., 2005; Pineda et al., 2014), rely on two probability-based assumptions related to holdover





time, which are both supported by the general shape of frequency distributions (Fig. 1): LIWs with short holdover times occur more frequently, while long holdover times are relatively rare in LIWs. Thus, the shorter the holdover time, the higher the degree of confidence in the ignition time. In fact, the minimum holdover time criteria is based almost exclusively on this assumption (Wotton and Martell, 2005; Moris et al., 2020). Moreover, a maximum holdover duration must be applied to limit the possibility that a lightning event may occur near a wildfire simply by pure chance (i.e., wildfires with long holdover

times might actually be human-caused; Nash and Johnson, 1996; Dowdy and Mills, 2009; Hessilt et al., 2022).

Further, we noticed that studies using remote sensing data (satellite images) as a source of fire ignition data present slightly different frequency distributions of holdover time. In contrast to what is typically reported, in daily and hourly frequency distributions the first time interval (i.e., the first day and hour respectively) is not the most frequent one. This may be due to

limitations of satellite data, such as temporal resolution (i.e., revisit period), spatial resolution (i.e., pixel size), and omission errors caused by clouds and smoke that affect the spatial and temporal accuracy of fire data products (Veraverbeke et al., 2014). For instance, active fires are only detectable if they are large or intense enough in relation to the pixel extent during the time of the satellite overpass, and not hindered by atmospheric conditions such as cloud cover or heavy smoke. These limitations could delay the discovery time of LIWs retrieved from satellite data (Hessilt et al., 2022; Menezes et al., 2022;

Xu et al., 2022). This may explain the underestimation of the frequency of short holdover times, displacing the mode of frequency distributions towards longer holdover times in comparison with studies that use fire records reported by more traditional methods. As a result, non-ecological factors, such as the lower wildfire detection capacity characteristic of remote areas (Wotton and Martell, 2005) and satellite-based active fire products (Johnston et al., 2018), may affect the estimation of holdover times, confounding the association with other drivers such as climate, weather, vegetation and soil, which are

typically used to explain the longer holdover times found in boreal regions (Fig. 2).

Similarly, recent holdover time data derived with the help of LLS are known to be affected by several data and methodological issues (Müller et al., 2013; Schultz et al., 2019; Moris et al., 2020; Pineda et al., 2022). Problems with wildfire data (e.g., misclassifications, low detection effort), lightning data (e.g., low detection efficiency and location

accuracy of LLS), and the methodology to match wildfires and lightning (e.g., criteria and parameters applied to select the most probable lightning igniting LIWs) can influence the estimations of holdover time. Nevertheless, we expect that these uncertainties derived from non-ecological factors will decrease with the improvement of data quality and methodological approaches. Furthermore, current and future instruments to detect lightning from satellites, such as the Geostationary Lightning Mapper (GLM) of the Geostationary Operational Environmental Satellite (GOES) R series (Goodman et al., 2013)

and the Lightning Imager (LI) of the Meteosat Third Generation (MTG) satellite series (Dobber and Grandell, 2014), can be used as a complementary and alternative source of lightning data to ground-based LSS. On the other hand, holdover time data derived during the 20th century, before the application of LLS, are consistent with the most recent datasets. In summary, the broad differences in data and methodological aspects between studies may complicate direct comparisons of

holdover time datasets (Table 3). Since users of the database may only be interested in certain datasets, ancillary data should
facilitate filtering censored and non-censored datasets.

The potential applications of the database on holdover times are diverse. (1) The data can be utilized to obtain descriptive statistics and plots on holdover time in different regions across the world. (2) Future studies may use the database to corroborate and compare their own holdover time estimates. (3) The database and the original studies listed in it may offer a
guide to obtain holdover time data and illustrate their main issues according to the scientific literature. (4) Theoretical probability distributions can be fitted using our frequency data to add a temporal dimension to the calculation of probabilities of lightning striking the reported LIW ignition areas (Hunt et al., 2017). (5) Similarly, the frequency distributions can help researchers select a probability distribution for their holdover time parametric models to identify fine-scale drivers of holdover duration or predict future durations under diverse scenarios. (6) Exploration and inferential models on holdover
time at broad scales could be tested using this database in combination with other datasets (e.g., from remote sensing).

## 5 Data availability

The current version of the database on holdover times of LIWs is freely available from Zenodo at https://doi.org/10.5281/zenodo.7352172 (Moris et al., 2022) under a CC-BY-4.0 license. Feedback on the data and files by users are welcome. The database includes code for data loading, plotting and basic manipulation within the R statistical
environment. The data may be expanded and updated in the future.

## 6 Conclusions

The main significance of this database is to become the first publicly available, harmonized, and ready-to-use global source of holdover time data. The current version of the database allows users to download and explore 42 frequency distributions of holdover time built with data on more than 150,000 LIWs from 13 countries and different periods extending from 1921 to
2020. By facilitating the access and analysis of different datasets of holdover time data, this database may become a significant data source for those interested in studying LIWs. Future research on LIWs will likely generate new holdover time data. Potential contributors to the database are thus encouraged to contact the corresponding author to discuss arrangements for sharing data. We expect that the database will be utilized in different ways, helping to improve our limited understanding of the holdover phenomenon and its implications for the study and modeling of LIWs.

**Author contributions**

JVM, PAA, HGPH and DA designed the database. JVM constructed the database. JVM, MC, AD, TDH, RL, LSM, MMM, FJPI, GBP, NP, RCS, SA and BMW contributed with data and metadata to the database. All authors discussed the database. JVM conceptualized and wrote the first draft of the manuscript. All authors contributed to the final draft of the manuscript.

**Competing interests**

One author is member of the editorial board of journal Earth System Science Data. The peer-review process was guided by an independent editor, and the authors have also no other competing interests to declare.

**Acknowledgements**

TDH acknowledges support under the umbrella of the Netherlands Earth System Science Centre (NESSC), funded by the European Union's Horizon 2020 research and innovation programme under the Marie Skłodowska-Curie Grant Agreement No. 847504. FJPI acknowledges the sponsorship provided by Junta de Andalucía under grant number POSTDOC-21-00052 and La Caixa Foundation under agreement number 116517.

**Financial support**

This research has been supported by a postdoctoral fellowship funded by the Government of Asturias (Spain) through FICYT (grant number AYUD/2021/58534).

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
