# Peer review of "A global database on holdover time of lightning-ignited wildfires"

_Earth System Science Data, 2022_

## Author Comment (AC1)

**Reviewer 1**

The paper is dedicated to the description of the construction and structure of a global database of lightning-ignited wildfire (LIW) holdover times collected from a large number of studies during the last century in different regions. The investigation of LIWs characteristics, especially on the background of global climate change is a topic of highly relevance and interest. Moreover, this database is the first freely available and usable in its current format. Since this data set is the first one and unique, it is probably incomplete, but it can be assumed that it will be constantly updated in the future, of course. Even now, the database on holdover has diverse potential applications. This database undoubtedly may become a significant data source for researchers interested in studying LIWs.

We would like to thank the reviewer for the positive comments.

However, I have some points that are unclear for me.

1. May be, it is worth to exclude Table 3 and Table 4 from the text? They are already presented in the database. Or if the authors want to provide an example of data, probably it will be better to make the Tables shorter? However, this is just a suggestion, it is on the authors' decision.

We included Tables 3 and 4 because we want to give a good overview of the database to the readers of the paper, especially with Table 3. We think that some readers may find more reasons to download the database if some information provided in these tables is of interest for them. Therefore, we prefer to keep these two tables.

We thought carefully about what information to show in Tables 3 and 4. Table 3 is on the long side (43 rows), yet it should still fit on one page in the published final version. Table 4 only contains ten rows and can easily be included in the manuscript.

2. What concerns Table 4, as I understood non-censored data also could be derived from the censored data? For what purposes non-censored data are provided (in Table 4 and in the database, in general)? Moreover, non-censored data are not discussed and analyzed in the text. How one can use them?

Non-censored data (summarized in Table 4) cannot be derived from censored data (summarized in Table 3); however, the opposite is true. As explained on Lines 134-135 of the manuscript, non-censored data refer to estimated values of continuous holdover time (i.e. the exact estimated value of holdover time for each single LIW); for example, 1.55 hours is a non-censored value of a holdover time. We can bin these continuous values of holdover time in interval classes (e.g. a histogram) to obtain censored data. Nine censored datasets in our database were derived from the non-censored datasets, as we pointed out on Lines 135-136 of the manuscript.

We believe that both non-censored and censored data may be of interest to the research community, yet fewer non-censored datasets exist. We obtained 42 datasets of censored data versus only 9 datasets of non-censored data. Thus, the manuscript has a strong focus on censored data given that the current version of the database relies heavily on frequency distributions. Nonetheless, for some applications, non-censored data (continuous values) may be preferred over censored data (frequency distributions). We want to give to users of the database the opportunity to work with the datasets as they see this fit

for their applications. Furthermore, we hope to add more non-censored datasets to the database in the future.

In general, the article is well structured and clear, the language consistent and precise. The article is appropriate to support the publication of a data set.

Best regards

Thank you very much for reviewing our manuscript.

---

## Author Comment (AC2)

**Reviewer 2**

This study developed a database on holdover time of lightning-ignited wildfires (LIWs) based on 29 studies through a literature review. The manuscript provided a summary of the proposed database. Holdover time is a valuable parameter to understand the behavior of LIWs. However, there are a few questions in this manuscript to be solved.

1. The holdover time and the location of each lightning-ignited wildfire are more valuable than the frequency distributions. Actually, readers can use holdover time and locations to calculate frequency by themselves and do more analysis. Can you provide the accurate holdover time and location of each LIW?

We agree with the reviewer that the holdover time and the location of each LIW are valuable information. However, we are not able to provide accurate holdover time and location of all single LIW because most of this information has not been published in the original documents we used to retrieve the data of our database or cannot always be shared. As we pointed out in the first paragraph of the discussion (L 248-251), the database does not include coordinates and dates of wildfires and lightning.

Some of the original studies used in the database were published in the previous century and it would be very challenging or impossible to obtain the raw data used to generate the holdover time values and distributions. In addition, some authors of more recent studies do not have permission to provide location data. As a result, we could not have built such as large database including detailed data on location.

Instead, in this manuscript we focus on a specific issue of LIW, i.e., holdover times. Given the lack of a comprehensive understanding about this phenomenon, a major goal of this database is to help researchers studying LIW by informing about the diversity of data and methods used to estimate holdover times, which is an important issue that researchers must face when combining lightning and wildfire data. We consider that these holdover time data are valuable and can provide important information for a wide range of studies. If we had access to all wildfire (and lightning) location and time data used in the original studies, the database would probably be very different (e.g., a spatially and temporally explicit LIW database instead of a holdover time database) and have more ambitious goals, such as describing diverse aspects of natural fire regimes.

2. Some wildfires didn't go to the arrival stage, and their holdover time was missing. Can you remove the biases in this database?

Wildfire data come usually from forest fire agencies, and these agencies are mainly interested in LIW that reach the arrival stage. We are aware that some LIW can extinguish during the survival phase (L 61-63), but to our knowledge there is no published research that attempted to study these particular LIWs, and so we do not know how often they occur, their variability among regions, drivers, etc.

Nevertheless, according to the definition of holdover time used for our database (i.e., the time between lightning-induced fire ignition and fire detection), these particular LIWs do not have a holdover time because they were never detected. Consequently, these LIWs are excluded of the database by definition. In fact, the durations from ignition to extinction of these LIWs may follow a different

distribution in comparison with holdover times of LIWs that clearly reached the arrival phase and were properly reported.

In conclusion, LIW that did not go to the arrival stage are out of the scope of the database. We added a sentence to the introduction (L 75) to make this clearer:

"On the contrary, those LIWs that extinguish before being detected or reaching the arrival phase are not considered to have a holdover time according to the definition presented above."

3. Figure 1a and Figure 1b showed relative frequencies from different study areas, i.e., Australia, Arizona and New Mexico. It is very difficult to compare the differences between daily and hourly frequency distributions because the study areas were also different. It is necessary to use the same study area.

The main objective of Figure 1 is to graphically highlight two different temporal resolutions of censored data present in the database (L 171-172, L 194-195), since many datasets have daily data (e.g., Austria in Figure 1a), while other datasets have hourly data (e.g., Arizona and New Mexico in Figure 1b). If we display the same dataset using different temporal resolutions, we will miss our objective with this figure. We also took advantage of Figure 1 to point out some properties of holdover time distributions throughout the manuscript, such as right skewness (L 226), most frequent time intervals (L 231 and 234), general shape (L 256), daily cycles (L 257-258), etc.

We also want to stress that the file "6_reproducible_examples.html" of the database (https://doi.org/10.5281/zenodo.7352172) includes several reproducible examples to work within the R statistical environment (L 189). Users can apply these examples to make their own quantitative and graphical comparisons.

4. In this database, some holdover time was shown by day while others were by hour, which was not consistent. Can you convert all data to the hourly scale?

Unfortunately, we cannot convert all data to the hourly scale. For instance, in daily censored data we only know the number of LIWs per 24-h interval, but we do not know how the holdover times of these fires are distributed per hour each day. We can convert hourly data to daily data, but not the opposite. However, we believe that presenting the censored data in the most accurate format (e.g., hourly when possible) is more beneficial than converting all data to days. For each dataset, we included the shortest time periods available from the original studies.

In the file "6_reproducible_examples.html" of the database (https://doi.org/10.5281/zenodo.7352172) we added examples showing how to aggregate censored and non-censored data. It is up to users of the database to aggregate or rescale the data according to their needs.

5. Figure 2, the biome 'Mediterranean forests' is not accurate. In Olson et al. (2001), it was 'Mediterranean forests, woodlands and scrub'.

We shortened the name of the Mediterranean biome in the database because we want to avoid commas in the csv file of ancillary data. Commas inside a field of a csv file are a potential source of problems that can break the format when csv files are read by a software. However, we took note and we will

solve this when we upload to Zenodo a new version of the database, probably by using other notation such as semicolons or underscores. Meanwhile, we changed the name of the biome in Figure 2: "Mediterranean forests, woodlands and scrub".

6. Table 3 showed very ambiguous study areas, e.g., California, which made it not user-friendly because there were different climate zones and vegetation distributions in California. Readers need more detailed locations. There were also study areas like Finland and Switzerland in Table 3, which were too coarse to use. Readers can't use this data if they want to establish the relationships between holdover time and climate, vegetation, terrain, etc.

Because generally we did not work with the exact locations of individual LIWs, we cannot divide all datasets from large study areas into different climate zones, vegetation classes, altitude ranges, etc, unless such divisions were done in the original research. This is the case of British Columbia, which was divided into two different zones: "WOT2022CA01" and "WOT2022CA02". Therefore, we would need the coordinates of single LIWs to carry out further divisions of the datasets. Nonetheless, this was not the scope of the database (see our reply to your comment #1).

As a result, researchers cannot use the database to establish relationships between holdover time and meteorology, vegetation, etc at a single LIW level (e.g., Pineda et al., 2022). However, the database may be useful to study broad-scale relationships of holdover time using the single "Study_id" (Table 3) as the unit of analyses, similar to the approach presented in Figure 2 (L 314-315). Other potential applications of the database are described briefly in the last paragraph of the discussion (L 307-315).

Pineda, N., Altube, P., Alcasena, F. J., Casellas, E., San Segundo, H., and Montanyà, J.: Characterizing the holdover phase of lightning-ignited wildfires in Catalonia, Agr. For. Meteorol., 324, 109111, https://doi.org/10.1016/j.agrformet.2022.109111, 2022.

7. Please add citations and more explanations to the maximum proximity index and the minimum holdover time on line 205.

Done. We added a short explanation and some references regarding these two methods in the introduction (L 97). The new sentence is the following:

"For example, the method based on the minimum holdover time selects the lightning event providing the shortest holdover duration (Wotton and Martell, 2005; Moris et al., 2020), while the method based on the proximity index developed by Larjavaara et al. (2005) selects the lightning event with the highest value of spatio-temporal proximity (Pineda et al., 2014)."

8. The diversity of the database is small. There were only 5 biomes in this study, where temperate broadleaf and mixed forests and flooded grasslands and savannas covered only two samples and one sample, respectively. However, there were 14 biomes globally. It is necessary to add more data.

We agree. The biogeographical diversity of the database is relatively small. This is due to the fact that so far most of the English scientific literature on LIWs came from a few regions. We acknowledge this limitation in the discussion (L 245-246). Our plan is to add more data in the future, which will depend on new publications on LIWs using holdover time data. Hopefully, we will be able to obtain data from new study areas and different biomes.

Best regards

We would like to thank the reviewer for the useful comments and suggestions.